# Long-Term Salinity-Responsive Transcriptome in Advanced Breeding Lines of Tomato

**DOI:** 10.3390/plants14010100

**Published:** 2025-01-01

**Authors:** Monther T. Sadder, Ahmad Abdelrahim Mohamed Ali, Abdullah A. Alsadon, Mahmoud A. Wahb-Allah

**Affiliations:** 1Department of Horticulture and Crop Science, School of Agriculture, University of Jordan, Amman 11942, Jordan; 2Department of Plant Production, College of Food and Agricultural Sciences, King Saud University, P.O. Box 2460, Riyadh 11451, Saudi Arabia; abdelrahima@estidamah.gov.sa (A.A.M.A.); alsadon@ksu.edu.sa (A.A.A.); 3The National Research and Development Center for Sustainable Agriculture, Riyadh Technology Valley, King Saud University, P.O. Box 2460, Riyadh 11451, Saudi Arabia; 4Vegetable Crops Department, Faculty of Agriculture, Alexandria University, Alexandria 21545, Egypt; abady625@hotmail.com

**Keywords:** *Solanum lycopersicum* L., salinity stress, RNA sequencing, gene expression

## Abstract

Soil salinity and the scarcity of freshwater resources are two of the most common environmental constraints that negatively affect plant growth and productivity worldwide. The tomato (*Solanum lycopersicum* Mill.) plant is moderately sensitive to salinity. The identification of salinity-responsive genes in tomato that control long-term salt tolerance could provide important guidelines for its breeding programs and genetic engineering. In this study, a holistic approach of RNA sequencing combined with measurements of physiological and agronomic traits were applied in two advanced tomato breeding lines (susceptible L46 and tolerant L56) under long-term salinity stress (9.6 dS m^−1^). Genotype L56 showed the up-regulation of known and novel differentially expressed genes (DEGs) that aid in the salinity tolerance, which was supported by a high salt tolerance index (81%). Genotype L46 showed both similar and different gene families of DEGs. For example, 22 paralogs of *CBL-interacting kinase* genes were more up-regulated in L56 than in L45. In addition, L56 deployed more *SALT OVERLY SENSITIVE* paralogs than L45. However, both genotypes showed the up-regulation of ROS-detoxifying enzymes and ROS-scavenging proteins under salinity stress. Therefore, L56 was more effective in conveying the stress message downstream along all available regulatory pathways. The salt-tolerant genotype L56 is genetically robust, as it shows an enhanced expression of a complete network of salt-responsive genes in response to saline conditions. In contrast, the salt-susceptible genotype L46 shows some potential genetic background. Both genotypes have great potential in future breeding programs.

## 1. Introduction

The most recent assessment report covering salinity was issued by the FAO [1]. Around 1.4 billion hectares (10.7%) of total global land area are affected by salinity, with an additional 1 billion hectares at risk due to the climate change and human mismanagement [1]. Salinity limits crop development and production [2]. One of the major affected crops is tomato (*Solanum lycopersicum* L.), which is a member of the Solanaceae family and is an annual horticultural crop with a wide distribution and high nutritional and economical value [3,4,5]. As cultivated tomato is moderately sensitive to salinity, all developmental stages are negatively affected [6,7,8].

Rapid fluctuation (up- and down-regulation) in stress-responsive genes is well established for several plant species including tomatoes [9]. The products of these genes cause signal transductions that lead to biochemical, physiological, and morphological changes involved in the final adaptation [9,10,11]. The transcriptome profiling of important regulatory genes is emerging as an important tool to improve plant response to abiotic stresses [12]. These genes include transcription factors, stress sensors or protein kinases that regulate the expression of several target genes for osmolyte biosynthesis enzymes, antioxidant enzymes, and stress proteins such as late embryogenesis abundant (LEA) proteins [13,14]. For the identification of differentially expressed genes, many advanced molecular biology techniques have been used to measure gene expression levels and discover new genes that are responsible for plant responses to different environmental stresses, such as quantitative real-time PCR (qPCR) [7,9] and transcriptome profiling using RNA sequencing [15].

Under abiotic stress, the signal perception of osmotic stress starts earlier than ionic stress, and the plant activates the signal transduction pathways by the generation of secondary messengers such as hormones, inositol, reactive oxygen species (ROS) and early responsive to dehydration (ERD) genes [2,16]. These molecules can stimulate signaling pathways such as the Ca^2+^-dependent salt overly sensitive (SOS) pathway and osmotic/oxidative stress signaling [6,8,15]. Under stress conditions, plant respiratory metabolism increases. Cytochrome c oxidase appears to play an active role in the electron transfer pathway during respiration within the mitochondria to produce a large amount of free energy (ATP) [17].

Recent studies have reported that calcium-dependent protein kinase (CDPK) is a part of the physiological plant defense mechanisms against biotic or abiotic attacks [18]. One of the most important transcription factors for abiotic stress responses in plant are ethylene-responsive factors (ERFs) [19]. The ERF gene family is part of the AP2/ERF superfamily. The AP2/ERF domain plays a major role in the gene expression regulation of other stress-responsive genes because it is highly conserved and binds to the GCC box present in the promoter region of stress-responsive genes [20]. Myloblastosis (MYB) [9] and (WRKY) [14] transcription factors are major factors in regulating plant metabolism and responses to abiotic stresses. The NAC domain (NAM, ATAF1 and 2, and CUC2) proteins are plant-specific transcription factors that have crucial roles in plant development and abiotic stress responses [9,21]. The plant cellular genome is adversely affected by oxidative, salt, and drought stresses and plants require the activation of the DNA repairing system.

The aim of the current study was to compare contrasting advanced breeding lines of tomato under long-term salinity stress (55 days), which has not been studied in previous articles. It is proposed that novel differentially expressed genes would be resolved between the two genotypes, which would generate applicable resources for future plant breeding.

## 2. Results

### 2.1. Agronomic Traits

A total of 19 important agronomically traits in tomato were assessed for two genotypes, L46 (salt-susceptible) and L56 (salt-tolerant). They cover vegetative growth traits (5), yield component traits (3), chemical traits (4), physiological traits (2) and fruit quality traits (4). The obtained data are hierarchically clustered and are illustrated as a heat map (Figure 1). Five major clusters of agronomic traits were visible. The first one clustered plant health, stem diameter, leaf area, leaf fresh and dry weights, indicating their tight correlation. The values of these traits gradually declined with an increasing salinity stress level in the susceptible genotype L46. On the contrary, the tolerant genotype L56 barely showed any decline with increasing salinity stress levels. The second group clustered leaf K^+^ and fruit ascorbic acid levels, where L56 showed a slight decline in the leaf K^+^ level at the highest salinity stress level (9.6 dS m^−1^), while L46 showed a decline in leaf dry weight with increasing salinity stress levels. Moreover, ascorbic acid levels showed a decline with increasing salinity levels in both genotypes. The third cluster included fruit fresh weight, average fruit weight, total yield and WUE, all of which were negatively affected with increasing salinity stress levels; however, this effect was more dramatic in the L46 genotype than in the L56 genotype. The reduction in total yield was a consequence of reduced fruit weight. Cluster IV had four traits, fruit number per plant, fruit pH, fruit TSS and leaf Ca^2+^, which were the least affected by salinity stress treatments. Cluster V included leaf proline, Na^+^ and Cl^−^ levels. Both Na^+^ and Cl^−^ levels were the most influenced by increasing salinity stress levels, showing a directly proportional relationship with increasing salinity stress levels and for both genotypes. On the other hand, leaf proline content was slightly decreased in the L56 genotype, while it was elevated in the L46 genotype. The K^+^/Na^+^ ratio did not cluster with any other trait and was the most negatively affected with increasing salinity stress levels, showing an inversely proportional relationship in both genotypes.

As a major index for plant response to salt stress, the salt tolerance trait index (STTI) was calculated for a group of nine major agronomic traits (Figure 2). At the 2.4 dS m^−1^ salinity stress level (the lowest level), both genotypes L46 and L56 were barely affected, showing relatively high STTI values (93–100%). At the second stress level (4.8 dS m^−1^), the L46 genotype showed a drastic drop in the STTI for the investigated traits (74–86%), while the L56 genotype was still showing high STTI levels (82–95%). At the third salinity stress level (7.2 dS m^−1^), severe plant damage was visible in the L46 genotype, with low STTI levels (56–80%). On the other hand, the L56 genotype showed two trends in STTI values, where plant growth parameters (plant height, stem diameter, leaf area, leaf fresh and dry weights) were marginally affected, with relatively high STTI levels (90–95%), while yield traits (fruit flesh thickness, average fruit weight and total yield) besides WUE were more negatively affected, with moderate STTI levels (79–82%). At the highest stress level (9.6 dS m^−1^), agronomic traits in L56 genotype showed two trends that were similar to the trends it showed at a lower stress level (7.2 dS m^−1^), with plant growth parameters showing higher STTI levels (85–93%) than yield traits and WUE (70–76%). Likewise, two major trends were clear for the L46 genotype at the highest salinity stress level (9.6 dS m^−1^), where plant growth parameters were strongly affected, with around 70–73% STTI levels, while yield traits and WUE were more strongly affected, with down to 50–54% STTI levels. The combined salt tolerance index (STI) revealed a huge significant difference between the L46 and L56 genotypes, with 61% and 82% STI values, respectively.

### 2.2. RNA Sequencing and Transcriptome Data Analysis

The aim of this part of the study was to investigate transcriptome changes in two tomato genotypes, L46 (salt-susceptible) and L56 (salt-tolerant), in response to salt stress. The cDNA libraries were synthesized for both tomato genotypes under the control (_c)/non-saline (0.5 dS m^−1^) condition and under the stress (_s)/salinity (9.6 dS m^−1^) condition. The delivered average data (three replicates) were around 4, 1.6, 2.3 and 5.4 million reads for L46_c, L46_s, L56_c and L56_s, respectively. Illumina reads passed quality filtering, with around 5% of the total reads trimmed at the 92 bp read length; the percentage of ambiguous nucleotides was less than 1%, which confirmed the accuracy of the sequencing. The PCA of all four samples revealed adequate proximity for all biological replicates in each sample (Appendix A).

The analysis of differentially expressed genes (DEGs) is an invaluable method to identify genes that may be responsible for salinity tolerance in the tomato L56 genotype. Therefore, DEGs were identified from the transcriptome data, with a *p* value ≤ 0.05 (Figure 3).

In the case of the L46_s vs. L46_c pair comparison, a total of 5732 DEGs were identified; 5228 with up-regulated expression and 504 with down-regulated expression (Figure 3A). However, with a threshold of Log2FC ≥ 2, only 1995 DEGs were resolved; 1781 with up-regulated expression and 214 with down-regulated expression (Figure 3B). On the other hand, in the L56_s vs. L56_c pair comparison, a total of 6186 DEGs were identified; 5544 with up-regulated expression and 642 with down-regulated expression (Figure 3C), while with a threshold of Log2FC ≥ 2, only 2475 transcripts showed significant changes (2255 up-regulated and 220 with down-regulated genes) (Figure 3D).

A comprehensive group of Venn diagrams were compiled for all possible combinations of the L56 and L45 genotypes and at a cutoff of fold changes ≥ 2, 4, 6, 8 and 10 for up-regulated DEGs (Appendix A). When L46_c was used as the baseline, L46_s showed an up-regulation of 1129, 465, 270, 91 and 52 unique DEGs with fold changes ≥ 2, 4, 6, 8 and 10, respectively. With the same baseline (L46_c), unique DEGs were also evident for L56_c at all fold levels. On the other hand, when L46_s was used as the baseline, L56_s showed an up-regulation of 3006, 972, 474, 278 and 181 unique DEGs with fold changes ≥ 2, 4, 6, 8 and 10, respectively. Because L56 is more tolerant to salinity compared with L46, the salt-responsive genes of L56 that are also differentially expressed in the L56_s vs. L46_s comparison (Appendix A) are more likely to play roles in the salinity tolerance of L56. Finally, when L56_c was used as the baseline, L56_s showed an up-regulation of 2790, 1690, 1080, 735 and 459 unique DEGs with fold changes ≥ 2, 4, 6, 8, and 10, respectively. When using any control genotype (L46_c or L56_c) as a baseline, a large number of DEGs were found to be up-regulated in the overlap between the investigated combination; 1797 and 1196 overlapped genes when L46_c and L56_c were used as the baseline, respectively (Appendix A).

Several thousand DEGs during salinity stress showed significant up-regulation or down-regulation (*p*-value < 0.01, fold change > 2 or <0.5). On the basis of similar kinetic patterns of expression, all DEGs were classified into a total of fifteen unique gene expression patterns (clusters) (Figure 4). Differentially expressed genes (DEGs) of the susceptible (L46) and tolerant (L56) genotypes under both control (_c) and stress (_s) conditions were used to reveal these 15 clusters. Each cluster was built from a large number of DEGs, ranging from 250 up to 1201 genes.

The gene expression patterns of cluster 1 (1192 DEGs), cluster 7 (488 DEGs), cluster 8 (566 DEGs) exhibit similar changes, where DEGs are up-regulated in L56 compared to L46 (under both control and salinity stress). The major difference between them is the fold change, which is minimal in cluster 1, moderate in cluster 7 and prominent in cluster 8 (Figure 4). On the contrary, cluster 5 (954 DEGs) shows the opposite pattern, where DEGs are up-regulated in L46 compared to L56 (under both control and salinity stress). Moreover, the gene expression patterns of cluster 2 (915 DEGs) and cluster 14 (533 DEGs) exhibit similar changes, where DEGs are up-regulated under salinity stress but with higher levels in the L56 genotype. Clustering DEGs in a similar pattern revealed major salinity-responsive genes, with a unique expression pattern in the tolerant genotype compared to the susceptible genotype. This was evident in cluster 3 (464 DEGs), cluster 11 (1201 DEGs), cluster 12 (317 DEGs) and cluster 15 (420 DEGs). Although the gene expression patterns exhibited diverse changes in these clusters, DEGs were largely up-regulated in the L56 genotype under salinity stress (Figure 4). On the contrary, both cluster 9 (394 DEGs) and cluster 13 (250 DEGs) showed the most down-regulated genes in the L56 genotype under salinity stress. On the other hand, cluster 6 (714 DEGs) showed highly expressed genes (up-regulated) with similar levels between both tolerant and susceptible genotypes.

A total of 22 *CBL-interacting kinase* genes were detected in the RNA-seq data. Their expression (FRKM calibration) were subjected to hierarchal clustering for L46_c, L56_c, L46_s and L46_s, and their expression levels are presented as a heat map (Figure 5). A major group of *CBL-interacting kinase* genes was up-regulated under salinity stress in both lines, with higher levels for some genes in the tolerant line (L56).

## 3. Discussion

### 3.1. Agronomic Traits in Response to Salinity Stress

It is very important to point out the most responsive group of traits (cluster V) in response to elevated long-term salinity stress in tomatoes (Figure 1). Cluster V included leaf proline, Na^+^ and Cl^−^ levels, which reflect major critical differences differentiating susceptible tomatoes from tolerant tomatoes. Nonetheless, the STTI was a very helpful index for plant response to salt stress (Figure 2), where you need to verify the tolerance with more clear phenotypic traits.

Most commercially available tomato cultivars are considered moderately sensitive to salinity stress. This was evident from norms of reaction for genotype L46 (susceptible compared to the tolerant genotype L56), placing it in the “moderately sensitive” division (Figure 6A,B). Tomato is classified as a glycophyte concerning response to salinity stress [7,9,22], which is mainly translated into osmolytes synthesis, e.g., proline [22] and sorbitol [23].

On the other hand, the L56 genotype (tolerant compared to the susceptible genotype L56) was placed in the “moderately tolerant” division (Figure 6A,C). This was true for both relative average fruit weight (Figure 6A) and total yield (Figure 6C), which reflects real reduction in crop production along elevated salinity levels. Salinity stress has a presumably less dramatic effect on relative leaf dry weight, which is an indirect contributor to crop production trait (Figure 6B).

Irrespective of classifications into glycophytes [5,23] or halophytes, relative crop production is the key feature to categorize any crop plant into a specific division in response to salinity stress [23,24]. Nonetheless, there are several agronomic traits that can be measured as a salinity stress index, which can be either directly (total yield) or indirectly (leaf dry weight) related to crop production.

### 3.2. Differential Gene Expression

Recent research in understanding salinity tolerance in different tomato genotypes at the molecular level have focused on identifying key genes and regulatory pathways that boost plant resilience. Above threshold levels, salinity induces oxidative stress in tomatoes, leading to ion imbalance and causing tremendous damage at the biochemical level [25,26].

Several reports have highlighted the role of antioxidants in mitigatoxidative damage, e.g., superoxide dismutase (SOD), catalase (CAT), and ascorbate peroxidase (APX) [27,28,29]. However, such studies were carried out on short-term exposure to salinity stress compared to the presented long-term study. Interestingly, ascorbic acid levels that should increase under stress conditions declined with increasing salinity levels in both tolerant and susceptible tomato lines in this investigations (Figure 1). These findings agree with similar studies carried out on tomatoes under salinity stress [30,31,32]. Nonetheless, ROS-mitigating genes were found to be up-regulated under long-term salinity stress compared to the control (Figure 7), which present a couple of each of the major ROS-detoxifying enzymes and ROS-scavenging proteins (superoxide dismutases, catalases, ascorbate peroxidases, glutathione peroxidases, peroxiredoxins, thioredoxins and glutaredoxins).

Moreover, neither the expression of transcription factors (e.g., *NAC* and *WRKY*) [9,33] nor ion homeostasis genes (e.g., *SOS*, *HKT1* and *NHX1*) [29] are enough to confer salinity tolerance for long-term stress exposure; even signaling pathways, e.g., cysteine-rich receptor-like protein kinases [34] or *CBL-interacting kinases* (this study), cannot, but rather, an array of collective genes and interactions from all levels are required as is shown in the comparative transcriptome analysis in this study. In fact, the top DEGs in the tolerant genotype L56 under salinity stress (Appendix A) reveal both compiled and compelled array of gene families known to mitigate salinity stress in tomatoes including transcription factors (*WRKY*, *MYB*, and *NAC*) and stress signaling pathways (*calmodulin*, *RLK*, and *S/T kinases*) [3,4,5,6,7,8,9].

This was clear in the presented long-term salinity study when salinity-related signaling pathways were investigated (Figure 5), where, for example, the *CBL-interacting kinase* genes were found to be up-regulated in L46 (susceptible genotype) as much as in L56 (tolerant genotype) or even more, e.g., Solyc06g068450.1.1 and Solyc05g052270.1.1. However, they were presumably not as effective as the downstream steps were missing a major link in the regulatory pathway.

Likewise, *SALT OVERLY SENSITIVE* (*SOS*) genes showed astonishing profiles comparing the two genotypes under both conditions (control vs. salinity) (Figure 8). None of the tomato *SOS* genes (for a full description, see Appendix A) were able to offer any decent variation between the two contrasting genotypes. This in turn supports the plausible variation between downstream processes resulting in a reduced damaging effect of salinity-imposed ions (comparing susceptible vs. tolerant genotypes). The tomato locus Solyc01g005020 encodes a Na^+^/H^+^ antiporter (*SlNHX* or *SOS1*), which is important in maintaining electrostatic balance via controlling ions (particularly Na^+^) inside the cell (Figure 8). It can sequester excess sodium into vacuoles under salinity stress (Figure 1). Therefore, it keeps the cellular component undamaged and fully functional [35].

Transcriptome analysis, by RNA-Seq and gene expression estimation using qPCR techniques, has been successfully used to study transcriptome profiling in a wide range of plant species [8,9,11,12,36]. Expression patterns of salinity-responsive genes were investigated with qPCR for a group of tomato genes (Appendix A). The fold increase in the expression level was compared to that revealed by the tomato RNA-Seq, which showed comparable trends in expression.

In general, gene expression patterns are not easy to detect in a particular transcriptome analysis. You can detect a few precise trends in Arabidopsis across seasons [37], but they are not well defined under salinity, e.g., in crayfish [38] or maize seedling roots [36]. However, we could distinguish an array of 15 different gene expression patterns from our transcriptome data (Figure 4). This reveals the power of this analysis to detect major genes in a particular trend in the tolerant genotype.

## 4. Materials and Methods

### 4.1. Plant Material, Growth Conditions, and Salt Stress Treatment

Two tomato advanced breeding lines (salt-susceptible (L46) and salt-tolerant (L56)) were used in this study. These lines were produced through the tomato breeding program at the Vegetable Improvement Unit, the College of Food and Agricultural Sciences, King Saud University [4]. This study was conducted under greenhouse conditions at the Agricultural Research and Experiment Station in Dirab. The seeds of both genotypes were sown in JV7 Pellets (Jiffy Products International Bv, Zwijndrecht, Norway) under growth chamber conditions, with diurnal temperatures of 27 ± 1 °C (day) and 19 ± 1 °C (night). One-month-old seedlings were transplanted into soil in a fiberglass greenhouse. The soil used was non-saline (EC 2.0–2.8 dS m^−1^), calcareous (CaCO_3_ 25–30%), and sandy in texture, with pH 7.3–7.8. The air temperature in the greenhouse was set to approximately 26 ± 1 °C during the day and 20 ± 1 °C at night, and the relative humidity was maintained at 75 ± 2% for the entire growing season. Fertilization and other cultural practices were applied as recommended for commercial tomato production [39].

Salinity treatment comprised five water salinity levels of NaCl (0.5, 2.4, 4.8, 7.2 and 9.6 dS m^−1^) applied through a drip irrigation system, where the control (0.5 dS m^−1^) was irrigated with irrigation water without any addition of NaCl. These salinity levels were selected based on a previous study [3]. Salinity treatments were started 5 days after transplanting using water containers (1 m^3^ each) connected to a surface drip irrigation network. Each container was filled with water at one of the salinity levels. The desired salinity level in irrigation water was developed by mixing appropriate amounts of NaCl according to the electric conductivity of the water source and the level of salt stress treatment [3].

The experiment was performed as a split plot in a completely randomized block design with three replications. Irrigation treatments were randomly allocated to the main plots, whereas genotypes were arranged in the sub-plots. The planting distance was 40 cm and 100 cm between plants and lines, respectively.

### 4.2. Agronomic Traits

Two months after transplanting (55 days of continuous stress treatment) (Appendix A), random samples of three plants from each sub-plot (with three blocks, we had a total of 9 plants) were chosen to measure plant height, stem diameter and leaf area. Leaf samples were collected, washed in distilled water and dried at 70 °C in a forced-air oven until the weight became constant (48–72 h), and the dry matter contents were calculated.

Fruits were picked by hand at 2–4-day intervals. Total yield, average fruit weight (the total weight of all harvested fruits per plot divided by their number), fruit number per plant and fruit flesh thickness were recorded.

The dried leaf samples were ground and used to determine the leaf content of Na^+^, Ca^2+^, K^+^, and Cl^−^. An analysis for sodium (Na^+^) and potassium (K^+^) was carried out by flame atomic absorption using sulfuric acid/hydrogen peroxide-digested plant material [40]. Calcium (Ca^2+^) analysis was carried out by atomic absorption spectroscopy (AAS) at 422.7 nm. A simple developed method for the effective extraction of Cl^−^ from plant tissue was used based on hot water extraction [41].

Water use efficiency (WUE) was calculated according to the following formula: WUE (kg m^−3^) = Total fruit yield (kg ha^−1^)/applied water (m^3^ ha^−1^). Free proline was determined using the method of [42].

At harvest, fruit samples from both genotypes were collected, juiced, and filtered for measuring the fruit content of total soluble solid (TSS), ascorbic acid or vitamin C, and acidity [43]. The salt tolerance trait index (STTI) was calculated by dividing the value of traits under the stress condition by the value of traits under the controlled condition [44]. In addition, the salt tolerance index (STI) was calculated as the mean of STTIs.

### 4.3. RNA Isolation and cDNA Preparation

Two months after transplanting, plant leaf tissues from two genotypes (L46 and L56) at two levels were used; the control (c) with 0.5 dS m^−1^ and the highest salt stress treatment (s) with 9.6 dS m^−1^. Three biological replicates were taken for each sample. Total RNA from tomato leaves was isolated using the RNeasy Plant Midi Kit (Qiagen, Hilden, Germany) according to the manufacturer’s protocol. All materials were treated with RNase Away (RNase Away, Molecular Bio Products, San Diego, CA, USA) to avoid the degradation of RNA by RNase. The RNA (5 μg) was used for the subsequent preparation of cDNA for each sample using the SMARTer cDNA Synthesis kit (Clontech, San Jose, CA, USA). The reaction was performed in 0.2 mL nuclease-free PCR tubes (Axgen, Stanford, CA, USA) according to the manufacturer’s instructions. The tubes were placed in a thermal cycler (Veriti 96-well; Applied Biosystems, Singapore) according to the manufacturer’s protocol.

### 4.4. RNA Sequencing and Data Analysis of Transcriptome

The RNA sequencing was performed at Virginia Tech (Blacksburg, VA, USA). All samples were bar-coded using dedicated primer adapters (Illumina, San Diego, CA, USA). The amplified and coded libraries were run as single reads with 101 cycles in GAIIx (Illumina, San Diego, CA, USA). The RNA sequencing was analyzed using CLC Genomics Workbench v. 9.1. The RNA-seq analysis was performed in several steps: First, all reads of RNA sequencing and the reference cDNA with their annotation were imported into the program. The 92 bp reads were mapped with the reference tomato cDS with the mapping set as described; up to three mismatches were allowed. The minimum length of the fraction was 0.9, the minimum similarity fraction was 0.8, and the maximum number of hits for a read was 10. The expression levels based on the number of mapped reads for the L46 transcriptome under saline and control conditions (L46 S/C) were compared with each other as were those of the L56 transcriptome under saline and control conditions (L56 S/C). In addition, the expression levels of L56 and L46 transcriptomes under the saline condition were also compared (L56 S/L46 S). The majority of reads could be mapped uniquely to one location within the tomato reference (cDNA) sequence. To normalize for sequencing depth and gene length, reads per kilobase of exon model per million mapped reads (RPKM) were calculated. DEGs with a *p*-value ≤ 0.05 were selected. CLC Genomics Workbench (Version 9.5) uses Student’s *t*-test as a statistical model to calculate the *p*-values for each DEG for two or more conditions. In addition, the software automatically adjusts the *p*-values using the Benjamani–Hochberg method to control the false discovery rate (FDR).

### 4.5. Primers Design

Genes that were highly expressed according to the transcriptome analysis were selected for confirmation using qPCR (Appendix A). The primer pairs for these genes were designed using Vector NTI 10 (Invitrogen, Carlsbad, CA, USA) and the Sol Genomics Network “https://solgenomics.net/organism/Solanum_lycopersicum/genome/” (accessed on 10 June 2024). The features were set into the program for primers designed as follows: primer length, 22–28 bp; Tm (°C), 58–60; GC %, 60; amplicon product length, 150–200 bp; and 3′ ends, CG.

### 4.6. Estimation of Gene Expression Level Using qPCR

To verify the expression level of the potential tomato salt-tolerant genes, qPCR was used. All RNA samples of both stressed and control plants were diluted to 10 ng ΜL^−1^ in nuclease-free water and subjected to first-strand cDNA synthesis using the GoScript Reverse transcription system (Promega, Madison, WI, USA). The cDNA was synthesized in repeat reactions to obtain a sufficient volume for qPCR. The reaction was performed in 0.2 mL nuclease-free PCR tubes (Axgen, Stanford, CA, USA) according to the manufacturer’s instructions. The qPCR assays were performed in 96-well PCR plates (Applied Biosystem, Singapore) using the quantitative PCR master mix (Quantifast SYBR Green PCR Kit; Qiagen, Germany). All qPCR assays were performed in triplicate. The reaction conditions were as recommended by the manufacturer. Quantification was performed using the 2^−ΔΔCT^ method [7], and the data were normalized for the quantity of actin transcripts.

## 5. Conclusions

The presented findings are critical for breeding novel long-term salinity-tolerant tomato varieties by deploying both traditional and molecular techniques. Continued explorations into detailed gene expression and downstream regulatory pathways are crucial for capturing the entirety of any tolerance regulatory network, either in tomato or any other crop. Such an approach could hold promise for mitigating the pronounced impact of salinity on global tomato crop production as climate change is striking each and every growing season around the world.

## Figures and Tables

**Figure 1 plants-14-00100-f001:**
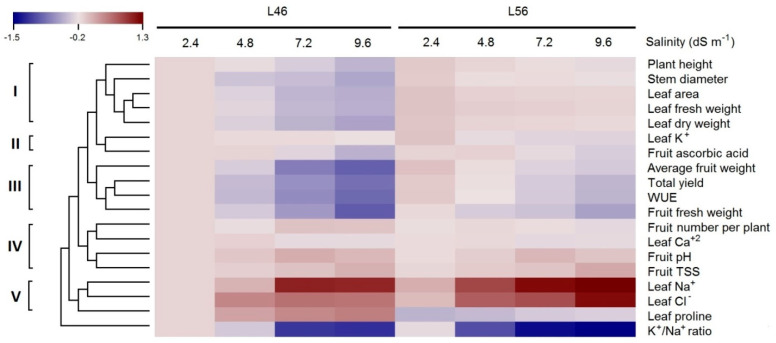
Hierarchical clustering and heat map of measured traits in susceptible (L46) and tolerant (L56) genotypes under four salinity stress levels (2.4, 4.8, 7.2 and 9.6 dS m^−1^). Each row represents a trait and each column represents a genotype under different stress levels. Five major clusters are evident (each with distinctive traits that differentiate them from other clusters).

**Figure 2 plants-14-00100-f002:**
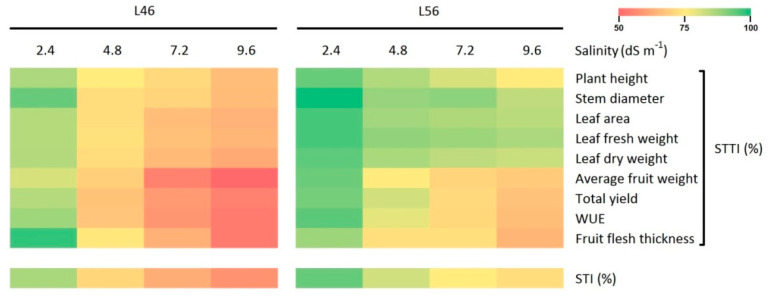
Heat map of salt tolerance trait index (STTI) for major agronomic traits in susceptible (L46) and tolerant (L56) genotypes under four salinity stress levels (2.4, 4.8, 7.2 and 9.6 dS m^−1^). The combined salt tolerance index (STI) for all traits is presented at the bottom of the figure. See Section 4.2 for STTI and STI calculations.

**Figure 3 plants-14-00100-f003:**
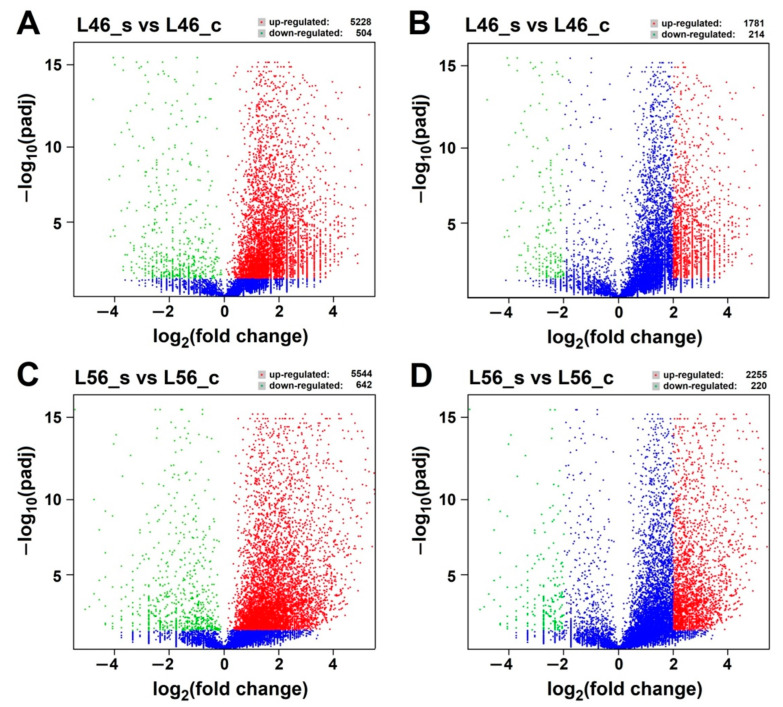
Global analysis of differentially expressed genes (DEGs). (**A**) Volcano plot of all DEGs for L46_s (stress) and L46_c (control) in transcriptome. (**B**) Volcano plot of DEGs (fold change ≥ 2) for L46_s (stress) and L46_c (control). (**C**) Volcano plot of all DEGs for L56_s (stress) and L56_c (control) in transcriptome. (**D**) Volcano plot of DEGs (fold change ≥ 2) for L56_s (stress) and L56_c (control). The abscissa shows the fold change difference in the expression of genes in different comparison groups, and the vertical coordinates indicate the adjusted *p*-values for the differences in expression. Genes without significant differences (*p* value ≤ 0.05) are indicated by blue dots below the threshold value (1.3). The up-regulated genes are represented by red dots, and the down-regulated genes are represented by green dots.

**Figure 4 plants-14-00100-f004:**
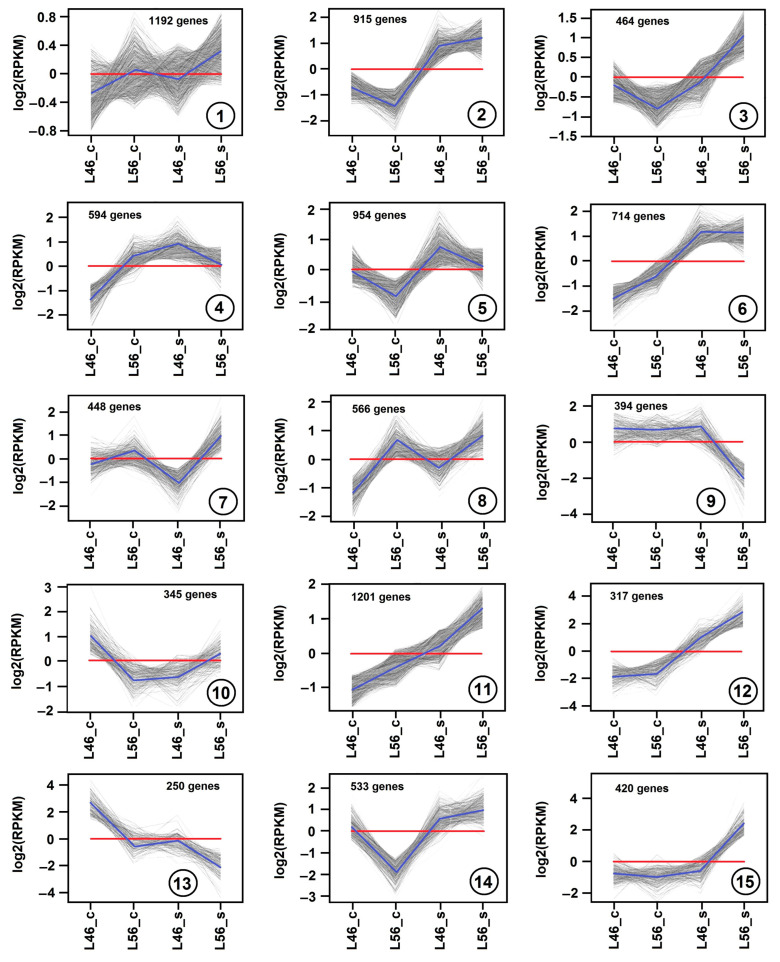
Gene expression patterns obtained by hierarchical clustering. Differentially expressed genes (DEGs) in tomato were categorized into 15 clusters (cluster number is depicted within a circle inside each panel). Gray lines show the relative expression levels of DEGs in the cluster in susceptible (L46) and tolerant (L56) genotypes under both control (_c) and stress (_s) conditions. Blue lines show the average values for each relative expression cluster. Red lines represent the baseline. Levels of gene expression were represented along the y axis as log2(RPKM), and genotypes were represented along the x axis.

**Figure 5 plants-14-00100-f005:**
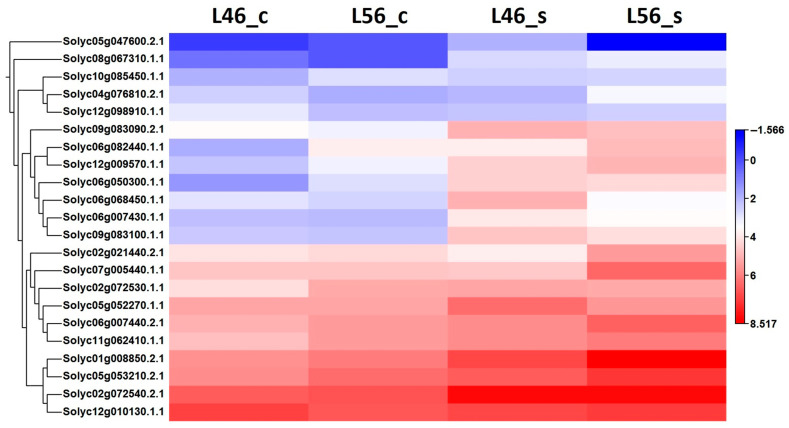
Hierarchical clustering of differentially expressed paralogs of *CBL-interacting kinases* of susceptible (L46) and tolerant (L56) genotypes under both control (_c) and stress (_s) conditions for all available stress-related genes in transcriptome.

**Figure 6 plants-14-00100-f006:**
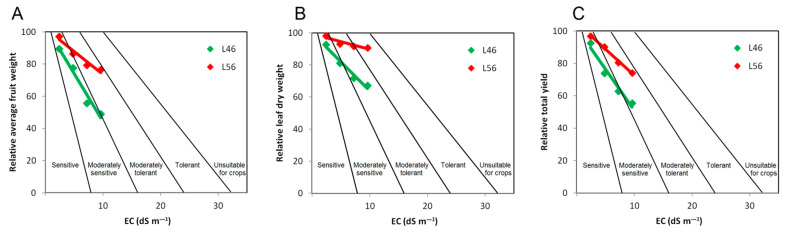
Norms of reaction for two tomato genotypes (L46 and L56) to different EC levels. Divisions represent plant salinity tolerance classes based on relative average fruit weight (**A**), leaf dry weight (**B**) and total yield (**C**) in comparison to control plants (modified from relative yield graph developed by [24]).

**Figure 7 plants-14-00100-f007:**
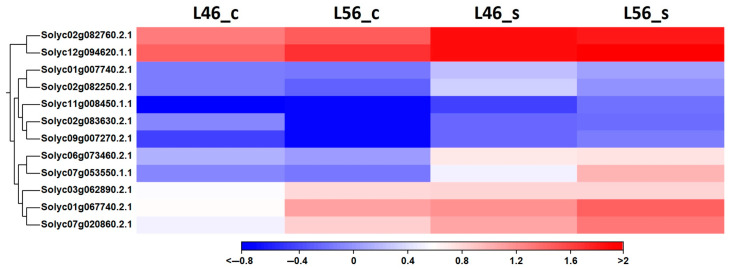
Hierarchical clustering of differentially expressed tomato ROS-detoxifying enzymes and ROS-scavenging proteins of susceptible (L46) and tolerant (L56) genotypes under both control (_c) and stress (_s) conditions for all available stress-related genes in transcriptome.

**Figure 8 plants-14-00100-f008:**
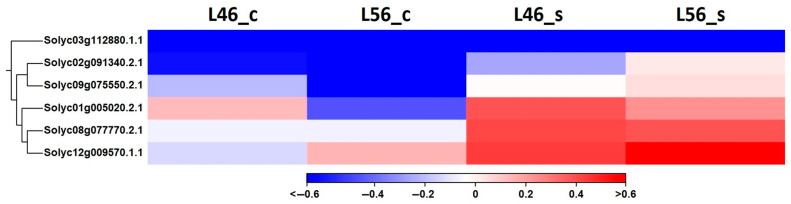
Hierarchical clustering of differentially expressed tomato paralogs for *SOS1-6* genes in susceptible (L46) and tolerant (L56) genotypes under both control (_c) and stress (_s) conditions for all available stress-related genes in transcriptome.

## Data Availability

RNA-seq data are available upon request. MTS AA AAA MAWA.

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
