# Peer review of "Long-Term Salinity-Responsive Transcriptome in Advanced Breeding Lines of Tomato"

_plants, 2025, doi:10.3390/plants14010100_

Round 1

Reviewer 1 Report

Comments and Suggestions for Authors

Dear Collegues

I read with great interest the manuscript received for review (Long-Term Salinity Responsive Transcriptome in Advanced 2 Breeding Lines of Tomato). The strength of this study is the attempt to combine agronomic, some physiological and molecular plant responses to salt stress to assess tolerance. The authors carried out extensive experimental work, validated the obtained transcriptomes, and attempted to combine agronomic traits and salt-regulated genes into clusters (groups) based on their response to stress. The main postulate of the authors is that the higher the DEG expression in response to salt stress, the better, the higher the salt tolerance of the plant. Unfortunately, this position is not uncontroversial.

Some comments

1- One of the main positions that the authors justify is that it is not one, two or three genes that determine salt tolerance, but a large number of them. From my point of view, this position does not require evidence, since all traits of plant resistance to abiotic factors, including salt tolerance, are polygenic in nature.

(2) The authors did not identify the biological function of DEGs that changed their expression level in response to salt stress, but only assessed the fold change in their transcripts and on this basis drew conclusions about their relation to salt stress (see Figure S2). The question arises, how was it determined whether a gene with altered expression intensity under salinization is relevant to the formation of salt tolerance in plants?

3.         Unfortunately, the authors did not provide any experimental data on the different salt tolerance of the compared lines.

4.         It would be good to give a clear statement of the objective of this study after the introduction, which is missing in the manuscript.

5.         I would recommend moving Figures S1 and S2 to the main text, as the data presented in them carry a significant scientific burden. It is not correct to depict RPKM levels as a curve in Figure S1.

6.         It is not clear from the methodological description how long the plants were exposed to salt stress? It would be interesting to know how salinity of the desired intensity was maintained using drip irrigation?

7.         It would be nice to express the salinity intensity not only in the conductivity of the solution, but also in the molar concentration of sodium chloride.

8.         A sample of 3 plants is definitely not sufficient for morpho-metric analyses.

9.         The text sometimes contains the following phrases: “Our findings are critical for breeding novel long term salinity tolerant tomato varieties by deploying both traditional and molecular techniques”. I would advise the authors to remove them.

Kind regards

Author Response

The authors are grateful for the review for the constructive comments and suggestions, which were addressed below with all made changes were highlighted in red in the manuscript.

Comment 1: [One of the main positions that the authors justify is that it is not one, two or three genes that determine salt tolerance, but a large number of them. From my point of view, this position does not require evidence, since all traits of plant resistance to abiotic factors, including salt tolerance, are polygenic in nature.]

Response 1: [You are absolutely right. The major issue is the differentiation between the tolerant and susceptible genotypes, where in the former one, additional genes and gene paralogs are being deployed to mitigate the stress and its devastating consequences on plant.]

Comment 2: [The authors did not identify the biological function of DEGs that changed their expression level in response to salt stress, but only assessed the fold change in their transcripts and on this basis drew conclusions about their relation to salt stress (see Figure S2). The question arises, how was it determined whether a gene with altered expression intensity under salinization is relevant to the formation of salt tolerance in plants?]

Response 2: [Thank you for this major comment. Additional tables showing top DEGs along with their description were added to the supplementary file and discussed in the manuscript.]

Comment 3: [Unfortunately, the authors did not provide any experimental data on the different salt tolerance of the compared lines.]

Response 3: [In fact all generated agronomical, physiological and biochemical data generated in this reveals the behavior of both lines (L46 and L56) under salinity stress (Figures 1 and 2).]

Comment 4: [It would be good to give a clear statement of the objective of this study after the introduction, which is missing in the manuscript.]

Response 4: [Highly appreciate this important missing point. It was added at the end of the introduction section.]

Comment 5: [I would recommend moving Figures S1 and S2 to the main text, as the data presented in them carry a significant scientific burden. It is not correct to depict RPKM levels as a curve in Figure S1.]

Response 5: [Thank you for these comments. You are right, however, the manuscript is loaded with other important figures, therefore, it would be better to keep them in the supplementary file. As for the RPKM, you are right, they were changed into dots without connecting lines.]

Comment 6: [It is not clear from the methodological description how long the plants were exposed to salt stress? It would be interesting to know how salinity of the desired intensity was maintained using drip irrigation?]

Response 6: [Very important piece of information, it was 55 days of continues stress treatment (starting 5 days after transplanting and ending 60 days after transplanting. This was clarified accordingly. For maintaining the stress levels, there were water containers (1 m3 each) connected to a surface drip irrigation network. Each container was filled with water at one of the salinity levels (section 4.1).]

Comment 7: [It would be nice to express the salinity intensity not only in the conductivity of the solution, but also in the molar concentration of sodium chloride.]

Response 7: [You are right but the source was having low concentration of mixed salts with 0.5 dS m-1, therefore, we kept the conductivity values as fixed. Nonetheless, it was less than 100 mM for the height level (9.6 dS m-1).]

Comment 8: [A sample of 3 plants is definitely not sufficient for morpho-metric analyses.]

Response 8: [You are right. As mentioned in section 4.1. we applied split-plot in randomized complete block design with three replications. And in section 4.2., random samples of three plants from each sub-plot. This means with three block, we had a total of 9 plants. This was added to section 4.2.]

Comment 9: [The text sometimes contains the following phrases: “Our findings are critical for breeding novel long term salinity tolerant tomato varieties by deploying both traditional and molecular techniques”. I would advise the authors to remove them.]

Response 9: [You completely right, we replaced all three positions containing "our" into neutral phrases.]

Reviewer 2 Report

Comments and Suggestions for Authors

The manuscript titled "Long-Term Salinity Responsive Transcriptome in Advanced Breeding Lines of Tomato" investigates the physiological and transcriptomic responses of two tomato genotypes under salinity stress. The study provides insights into salinity tolerance mechanisms through RNA sequencing and agronomic trait analysis. However, the manuscript requires significant revisions to enhance clarity, scientific rigor, and overall coherence. Key concerns include the lack of sufficient methodological details, incomplete comparisons with existing research, overstated conclusions about genotype superiority, significant gaps in the description of the results and technical data. Therefore, a major revision is recommended.

Abstract

-Correct "holist approach" to "holistic approach."

-Provide a more specific summary of the results, highlighting key findings and their implications.

Introduction

-Clearly state the issue of salinity in agriculture, including relevant statistics or observed trends to contextualize the problem.

-Indicate the novelty of the study in comparison to prior research, explaining its unique contribution.

-Propose a hypothesis that the study aims to test based on existing knowledge gaps.

Results

-Include photographs of plant genotypes under normal and stress conditions for visual support.

-Lines 75-77: This paragraph appears to be phrased in a way that resembles a prompt or request to AI.

-Lines 85-86: Correct the typo in the numbers of resistant and sensitive lines.

-Lines 75-77: The description of cluster 2 highlights potassium ion and ascorbic acid content as key features; however, the authors fail to compare these traits between the two genotypes across different salinity levels. Additionally, while differences in ascorbic acid content are clearly evident, they are not discussed in the text. .

-Clarify how, despite a reduction in fruit weight and number of fruits, the overall yield in L46 remains relatively unaffected.

-Lines 90 and 92: Clarify why the "fruit number per plant" trait is included in both clusters (3 and 4).

-Line 94: Replace "C-" with "Cl-."

-Line 95: Avoid using "positively affected" when discussing sodium and chloride ion accumulation. This term is misleading in this context.

-Discuss the strong sodium ion accumulation compared to the slight decrease in potassium ions for both lines.

-Salt stress induces the formation of reactive oxygen species (ROS) and is closely associated with lipid peroxidation, which significantly impacts cell integrity and function under stress. Including data on ROS levels and lipid peroxidation would strengthen the authors' hypotheses by providing direct evidence of oxidative damage and its mitigation in the studied genotypes. 

-Lines 114-121: Reorganize the logical description of results for better coherence.

-Some points are repeated across sections, such as the discussion of the role of STTI in assessing plant responses. These repetitions should be streamlined to improve clarity and conciseness. Consolidating the information into a single, well-structured section would provide a more cohesive understanding of STTI’s relevance and its findings without unnecessary redundancy.

-Lines 139-140: Explain how library size differences were normalized. Provide PCA or MDS data to demonstrate that technical biases were addressed.

-Include a table showing the number of trimmed reads and the remaining reads for transparency.

-Lines 143-145: Specify the statistical tests used to calculate p-values and fold changes.

-Line 168: Correct the genotype labeling to avoid confusion.

-Figure 4: Rearrange the graphs vertically to ensure full visibility.

-Line 169: Correct the reference to Figure S1, which should be Figure S2.

-Figure S1: Provide the rationale for selecting the nine genes analyzed and include their names or functions.

-Line 232: Replace "C-" with "Cl-."

Discussion

-The discussion mentions the importance of proline, Na+, and Cl- in leaves but lacks references or descriptions of their roles. Compare these results with previous studies to highlight new contributions.

-The authors provide a detailed account of the number of differentially expressed genes (DEGs) in the two genotypes. However, the description lacks depth in discussing the specific functions of these genes, their involvement in relevant pathways, and how they contribute to salinity tolerance. To strengthen the analysis, the authors should explore the biological roles of key DEGs, their interactions within regulatory networks, and their impact on physiological traits. 

-Avoid overly optimistic claims about L56's superiority without additional field validations or comparisons with other genotypes.

-Identify key genes or pathways that contribute to salinity tolerance. Discuss their potential applications and, if possible, include data from functional validation experiments (e.g., knockout or overexpression studies).

Materials and Methods

-Provide more details about biological and technical replicates and specify the number of RNA-seq replicates used to ensure robustness.

-Although the salinity levels (0.5 to 9.6 dS m⁻¹) are suitable, higher levels should be tested to explore tolerance limits.

-Provide detailed descriptions of the methods used for transcriptome analysis, validation, and pathway enrichment, as well as the functional roles of salinity-responsive genes.

-Conduct functional annotation and pathway analysis of DEGs to enhance the depth of the study.

Author Response

The Authors are grateful for the review for the constructive comments and suggestions, which were addressed below with all made changes were highlighted in red in the manuscript.

Comment 1: [Abstract -Correct "holist approach" to "holistic approach."]

Response 1:[Thank you for your comment. Correction was made]

Comment 2: [Abstract -Provide a more specific summary of the results, highlighting key findings and their implications.]

Response 2: [Thank you for this good suggestion. The abstract was improved accordingly.]

Comment 3: [Introduction -Clearly state the issue of salinity in agriculture, including relevant statistics or observed trends to contextualize the problem.]

Response 3: [Thank you for this comment. A 2024 FAO related report was added at the beginning of the introduction]

Comment 4: [Introduction -Indicate the novelty of the study in comparison to prior research, explaining its unique contribution.]

Response 4: [Thank you, it was added at the end of the introduction.]

Comment 5: [Introduction -Propose a hypothesis that the study aims to test based on existing knowledge gaps.]

Response 5: [Thank you, it was added at the end of the introduction.]

Comment 6: [Results -Include photographs of plant genotypes under normal and stress conditions for visual support.]

Response 6: [This is a good suggestion. A photograph showing the irrigation system and transplants in greenhouse was added to the supplementary file.]

Comment 7: [Results -Lines 75-77: This paragraph appears to be phrased in a way that resembles a prompt or request to AI.]

Response 7: [Thank you for the comment, this was available in the template used from the journal submission site. We did not apply any AI data at all. The sentence was removed.]

Comment 8: [Results -Lines 85-86: Correct the typo in the numbers of resistant and sensitive lines.]

Response 8: [Thank you for your comment, the line number was corrected.]

Comment 9: [Results -Lines 85-87: The description of cluster 2 highlights potassium ion and ascorbic acid content as key features; however, the authors fail to compare these traits between the two genotypes across different salinity levels. Additionally, while differences in ascorbic acid content are clearly evident, they are not discussed in the text.]

Response 9: [You are right, it is very interesting results, as usually ascorbic acid content increases with stress, however, it was declining in our case with increasing salinity levels. This result was added. Moreover, it was also mentioned in discussion as this was recorded in tomato exposed to salinity stress in multiple articles.]

Comment 10: [Results -Clarify how, despite a reduction in fruit weight and number of fruits, the overall yield in L46 remains relatively unaffected.]

Response 10: [In fact, total yield was decreasing in L46 as salinity stress was increasing (Figure 1). This fact was clarified in the results section.]

Comment 11: [Results -Lines 90 and 92: Clarify why the "fruit number per plant" trait is included in both clusters (3 and 4).]

Response 11: [You are right, it was added to cluster III in the text by mistake and was corrected]

Comment 12: [Results -Line 94: Replace "C-" with "Cl-."]

Response 12: [ Thank you for your comments and it was corrected.]

Comment 13: [Results -Line 95: Avoid using "positively affected" when discussing sodium and chloride ion accumulation. This term is misleading in this context.]

Response 13: [You are right, it is rather misleading, it not is positive for the plant but rather it reflects the increase in level. The sentence was corrected accordingly.]

Comment 14: [Results -Discuss the strong sodium ion accumulation compared to the slight decrease in potassium ions for both lines.]

Response 14: [Thank you, it was discussed as request.]

Comment 15: [Results -Salt stress induces the formation of reactive oxygen species (ROS) and is closely associated with lipid peroxidation, which significantly impacts cell integrity and function under stress. Including data on ROS levels and lipid peroxidation would strengthen the authors' hypotheses by providing direct evidence of oxidative damage and its mitigation in the studied genotypes.]

Response 15: [I agree with your important suggestion. Therefore, DEGs related to ROS detoxifying enzymes and ROS-scavenging proteins were presented and discussed.]

Comment 16: [Results -Lines 114-121: Reorganize the logical description of results for better coherence.]

Response 16: [Thank you for this comments. Those sentences were re-written in better and clearer way.]

Comment 17: [Results -Some points are repeated across sections, such as the discussion of the role of STTI in assessing plant responses. These repetitions should be streamlined to improve clarity and conciseness. Consolidating the information into a single, well-structured section would provide a more cohesive understanding of STTI’s relevance and its findings without unnecessary redundancy.]

Response 17: [Thank you for this comments. The STTI section was re-written in better and clearer way.]

Comment 18: [Results -Lines 139-140: Explain how library size differences were normalized. Provide PCA or MDS data to demonstrate that technical biases were addressed.]

Response 18: [Thank you for mentioning these points. Library size was calibrated using RPKM. And PCA analysis was included showing closely related samples.]

Comment 19: [Results -Include a table showing the number of trimmed reads and the remaining reads for transparency.]

Response 19: [The trimmed reads of around 5% was recorded for al sequenced libraries.]

Comment 20: [Results -Lines 143-145: Specify the statistical tests used to calculate p-values and fold changes.]

Response 20: [The requested data was added to section 4.4.; CLC Genomics Workbench uses Student’s t-test as statistical model to calculate the p-values for each DEGs between two or more conditions. In addition, the software automatically adjusts the p-values using the Benjamini-Hochberg method to control the False Discovery Rate (FDR).]

Comment 21: [Results -Line 168: Correct the genotype labeling to avoid confusion.]

Response 21: [Thank you, it was corrected]

Comment 22: [Results -Figure 4: Rearrange the graphs vertically to ensure full visibility.]

Response 22: [Good idea for better visibility, and it was rearranged vertically.]

Comment 23: [Results -Line 169: Correct the reference to Figure S1, which should be Figure S2.]

Response 23: [It was corrected accordingly.]

Comment 24: [Results -Figure S1: Provide the rationale for selecting the nine genes analyzed and include their names or functions.]

Response 24: [These genes were selected randomly to cover multiple gene expression patterns (Figure 4). Their description is available above (Supplementary Table S2). And this hint was added to the legend.]

Comment 25: [Results -Line 232: Replace "C-" with "Cl-."]

Response 25: [It was corrected accordingly.]

Comment 26: [Discussion -The discussion mentions the importance of proline, Na+, and Cl- in leaves but lacks references or descriptions of their roles. Compare these results with previous studies to highlight new contributions.]

Response 26: [Thank you for raining this point. The importance of Na+/H+ antiporter (SlNHX or SOS1) and its expression was discussed.]

Comment 27: [Discussion -The authors provide a detailed account of the number of differentially expressed genes (DEGs) in the two genotypes. However, the description lacks depth in discussing the specific functions of these genes, their involvement in relevant pathways, and how they contribute to salinity tolerance. To strengthen the analysis, the authors should explore the biological roles of key DEGs, their interactions within regulatory networks, and their impact on physiological traits. ]

Response 27: [You are absolutely right. The Description of major top DEGs were included and focus in major groups as SOSs, Kinases and major ROS detoxifying enzymes and ROS-scavenging proteins were presented and discussed.]

Comment 28: [Discussion -Avoid overly optimistic claims about L56's superiority without additional field validations or comparisons with other genotypes.]

Response 28: [You are right, However, we have field data for multiple years with multiple publications, and it is even was recently used as a reference salinity tolerant line.]

Comment 29: [Discussion -Identify key genes or pathways that contribute to salinity tolerance. Discuss their potential applications and, if possible, include data from functional validation experiments (e.g., knockout or overexpression studies).]

Response 29: [That was added to discussion section, however, the data show that it is no way a single gene or few genes effect, but rather an array of multiple layers of gene regulations with multiple paralogs in tomato, therefore, knockout or overexpression studies would be difficult to attain unless paralogs are targeted simultaneously, which would be a key research target in the future.]

Comment 30: [Materials and Methods -Provide more details about biological and technical replicates and specify the number of RNA-seq replicates used to ensure robustness.]

Response 30: [That was added as requested.]

Comment 31: [Materials and Methods -Although the salinity levels (0.5 to 9.6 dS m⁻¹) are suitable, higher levels should be tested to explore tolerance limits.]

Response 31: [We already investigated higher levels in earlier publications [3] , which was devastating to susceptible lines.]

Comment 32: [Materials and Methods -Provide detailed descriptions of the methods used for transcriptome analysis, validation, and pathway enrichment, as well as the functional roles of salinity-responsive genes.]

Response 32: [That was added in different location along the manuscript.]

Comment 33: [Materials and Methods -Conduct functional annotation and pathway analysis of DEGs to enhance the depth of the study.]

Response 33: [Thank you, again top DEGs were mentioned and additional major groups were discussed.]

Round 2

Reviewer 2 Report

Comments and Suggestions for Authors

The authors have improved the manuscript in response to my previous comments. However, several important corrections have not been addressed.

  1. My initial comment about including photographs of plant genotypes under normal and stress conditions for visual support has not been addressed. The provided photograph of seedling irrigation does not adequately capture the physical condition of the plants at the end of the salt exposure period. I strongly recommend including high-quality photographs of the two lines (L46 and L56) under both control and salinity stress conditions. These images should clearly depict the visible differences in growth, leaf condition, and overall plant health, offering a more comprehensive understanding of the phenotypic impact of salinity stress.

  2. The paragraph in lines 84–86, which the authors claim is a template used from the journal submission site, remains in place and has not been removed as requested. Please delete this section, as it is not appropriate for the manuscript.

  3. There are errors in genotype labeling throughout the text. Specifically, the genotype is incorrectly written as 65 instead of 56 in lines 95, 121, and 123. Please ensure that all genotype labeling is consistent and correct throughout the manuscript.

Author Response

The authors would like to thank the reviewer from the constructive comments on the revised manuscript. All new changes were made as required and were highlighted in red.

Comment 1: [My initial comment about including photographs of plant genotypes under normal and stress conditions for visual support has not been addressed. The provided photograph of seedling irrigation does not adequately capture the physical condition of the plants at the end of the salt exposure period. I strongly recommend including high-quality photographs of the two lines (L46 and L56) under both control and salinity stress conditions. These images should clearly depict the visible differences in growth, leaf condition, and overall plant health, offering a more comprehensive understanding of the phenotypic impact of salinity stress.]

Response 1: [That is clear, and a new image was added as requested showing the two lines.]

Comment 2: [The paragraph in lines 84–86, which the authors claim is a template used from the journal submission site, remains in place and has not been removed as requested. Please delete this section, as it is not appropriate for the manuscript.

Response 2: [Thank you again, and it was removed accordingly.]

Comment 3: [There are errors in genotype labeling throughout the text. Specifically, the genotype is incorrectly written as 65 instead of 56 in lines 95, 121, and 123. Please ensure that all genotype labeling is consistent and correct throughout the manuscript.

Response 3: [Thank you again for this comment. All line numbers were corrected accordingly.]